# Growth Hormone Secretory Capacity Is Associated with Cardiac Morphology and Function in Overweight and Obese Patients: A Controlled, Cross-Sectional Study

**DOI:** 10.3390/cells11152420

**Published:** 2022-08-04

**Authors:** Elena Gangitano, Giuseppe Barbaro, Martina Susi, Rebecca Rossetti, Maria Elena Spoltore, Davide Masi, Rossella Tozzi, Stefania Mariani, Lucio Gnessi, Carla Lubrano

**Affiliations:** 1Department of Experimental Medicine, Sapienza University of Rome, 00161 Rome, Italy; 2Department of Molecular Medicine, Sapienza University of Rome, 00161 Rome, Italy

**Keywords:** cardiovascular risk, echocardiography, epicardial fat, growth hormone, growth hormone deficiency, heart, metabolic syndrome, obesity

## Abstract

Obesity is associated with increased cardiovascular morbidity. Adult patients with growth hormone deficiency (GHD) show morpho-functional cardiological alterations. A total of 353 overweight/obese patients are enrolled in the period between 2009 and 2019 to assess the relationships between GH secretory capacity and the metabolic phenotype, cardiovascular risk factors, body composition and cardiac echocardiographic parameters. All patients underwent GHRH + arginine test to evaluate GH secretory capacity, DEXA for body composition assessment and transthoracic echocardiography. Blood samples are also collected for the evaluation of metabolic parameters. In total, 144 patients had GH deficiency and 209 patients had normal GH secretion. In comparing the two groups, we found significant differences in body fat distribution with predominantly visceral adipose tissue accumulation in GHD patients. Metabolic syndrome is more prevalent in the GHD group. In particular, fasting glycemia, triglycerides and systolic and diastolic blood pressure are found to be linearly correlated with GH secretory capacity. Epicardial fat thickness, E/A ratio and indexed ventricular mass are worse in the GHD group. In the population studied, metabolic phenotype, body composition, cardiovascular risk factors and cardiac morphology are found to be related to the GH secretory capacity. GH secretion in the obese patient seems to be an important determinant of metabolic health.

## 1. Introduction

Obesity is a multifactorial disease associated with increased morbidity and mortality, especially from cardiovascular events. A wide spectrum of metabolic and hormonal alterations is associated with obesity, and among these, reduced circulating growth hormone (GH) levels have been observed [1,2].

Adult patients with GH deficiency (GHD) have an increased cardiovascular risk related to metabolic syndrome (MS) and morpho-functional cardiological alterations [3].

The pattern of fat deposition can be altered in GHD, and visceral fat may be increased. Epicardial fat is a marker of visceral adiposity and cardio-metabolic risk [4,5] and can be easily and non-invasively determined through echocardiography.

Data on cardiac morphology and function in GHD patients with obesity or overweight are lacking. GHD can have direct effects on the heart, resulting in reduced cardiac output as well as impaired diastolic and systolic function; for this reason, it may act as an additive factor to the cardiovascular risk that is already increased in patients with obesity.

The purpose of this study was to evaluate the metabolic phenotype and the cardiac function and morphology in a population of patients suffering from obesity or being overweight in relation to their GH secretory capacity.

## 2. Materials and Methods

### 2.1. Study Design and Participants

This was a retrospective, cross-sectional, controlled study. Participants for this study were recruited among those who referred to the High Specialization Centre for the Care of Obesity (CASCO), in Umberto I Polyclinic, Sapienza University of Rome, between 2014 and 2019. Some patients were also included in other studies already published by our group [6,7].

Inclusion criteria were as follows: age between 18 and 65, body mass index (BMI) ≥ 25 kg/m^2^, serum insulin-like growth factor 1 (IGF-1) levels reduced or close to the lower limit of the normal range for age and sex and presenting at least one sign or symptom suggesting GH deficiency (osteopenia, sarcopenia, altered fat mass distribution, altered echocardiographic parameters, dyslipidaemia).

Exclusion criteria were as follows: pregnancy, breastfeeding, active neoplastic diseases, tobacco habit or alcohol abuse, current medication with drugs known to affect pituitary. Patients received medications to control hypothyroidism, diabetes mellitus type 2, dyslipidemia, hypertension, gastritis and depression when required.

All patients had their medical history collected and underwent a complete clinical, metabolic and cardiological evaluation with physical exam, laboratory exams and GHRH plus Arginine stimulation test as part of routine diagnostic workup during hospitalization. All patients gave their written informed consent. The study was approved by the Medical Ethical Committee of Sapienza University of Rome (ref. CE5475) and all procedures were performed in accordance with the Declaration of Helsinki (1964) and subsequent amendments.

### 2.2. Anthropometric Measurements

Anthropometric parameters were obtained between 8 and 10 a.m. in fasting subjects wearing light clothing and no shoes.

Body weight was rounded to the nearest 0.1 kg and height to the nearest 0.1 cm. waist circumference (WC) was measured with non-stretchable tape over the unclothed abdomen just above the iliac crest at the end of a normal expiration, and hip circumference (HC) was measured around the pelvis at the widest point. The tape was parallel to the floor and did not compress the skin.

Waist-to-Hip Ratio (WHR) was calculated using the parameters described above.

BMI was calculated as weight (kg) divided by squared height (m^2^). Overweight was defined as a BMI ≥ 25 kg/m^2^ and <30 kg/m^2^, whereas obesity as a BMI ≥ 30 kg/m^2^. Arterial blood pressure (BP) was measured with an aneroid sphygmomanometer in the right arm, with the patients in the sitting position after five minutes of rest.

### 2.3. GH Secretory Capacity Evaluation

IGF-1 was assayed by an immunoradiometric assay, after ethanol extraction (Diagnostic System Laboratories Inc., Webster, TX, USA). The normal ranges in <23, 23–30, 30–50, 50–100-year-old patients were 195–630, 180–420, 100–415, 70–250 mg/L respectively.

All patients presenting with reduced serum IGF-1 levels or close to the lower limit of the normal range for age and sex and presenting at least one sign or symptom suggestive of GHD underwent a dynamic stimulation test with GHRH + arginine. The exam was conducted administering an intravenous bolus injection of 1 µg/kg of GHRH and 0.5 g/kg (until a maximum of 30 g) of arginine hydrochloride in 30 min intravenous infusion. Blood samples were collected at −15, 0, + 30, +45 and +60 min. The results were interpreted using BMI-related cut-off values so that GHD was diagnosed when peak value was below 4.0 ng/dL in patients with obesity and below 8 ng/dL in patients with overweight [8]. The area under the concentration-time curve (AUC) was calculated using the trapezoidal rule to assess GH secretory capacity.

### 2.4. Biochemical Assessment

Blood samples were collected from fasting patients by venepuncture between 8 and 9 a.m. Samples were then transferred to the local laboratory and handled according to the local standards of practice. A complete metabolic assessment including fasting glucose, insulin, HbA1c, total cholesterol, high-density lipoprotein (HDL) cholesterol, triglycerides (TG) was performed. Insulin resistance was assessed with the HOMA-IR (homeostatic model assessment of insulin resistance), calculated as fasting insulin (UI/L) x fasting glucose (mg/dL)/405 and patients were defined insulin resistant when HOMA-IR exceeded 2.5. Metabolic syndrome (MS) was diagnosed according to the ATP III criteria (i.e., the presence of at least three among the following five features: WC ≥ 102 cm in men or ≥88 cm in women, BP ≥ 130/85 mmHg, TG level ≥ 150 mg/dL, HDL cholesterol level < 40 mg/dL in men or <50 mg/dL in women and fasting glucose ≥ 100 mg/dL)) [9].

### 2.5. Dual Energy X-ray Absorptiometry

All patients underwent dual-energy X-ray absorptiometry (DXA) (Hologic A Inc., Bedford, MA, USA, QDR 4500W, software version 12.5.3:2) to evaluate human body composition.

All scans were administered by trained research technicians using standardized procedures recommended by GE Healthcare. DXA was performed with subjects wearing light clothing and no shoes.

For each patient, the following parameters were measured: body fat or fat mass (FM), trunk fat and fat-free mass or lean mass (FFM), expressed in absolute value in Kg and percentage. The upper body fat deposition index (UFDI) was calculated as previously reported [10] as the ratio between upper fat (head, arms and trunk fat in Kg) and lower fat (legs fat in Kg). Appendicular skeletal muscle mass (ASMM) was calculated as the sum of arms and legs FFM, and the ratio ASMM (Kg)/weight (Kg) was utilized as a parameter of sarcopenia [11].

### 2.6. Echocardiography

Participants underwent high-resolution M-B mode trans-thoracic echocardiography with commercially available ultrasound systems, using a 2.5 MHz probe (Esaote MyLab40, Esaote Europe B.V., Maastricht, The Netherlands). The following parameters were assessed: left ventricle ejection fraction (EF, expressed in %), interventricular septal thickness (IVS, expressed in mm); indexed left ventricular mass (LVMi, expressed in g/m^2^); circumferential fractional shortening (cFS, expressed in mm); left ventricular end systolic diameter (LV ESD, expressed in mm); left ventricular end diastolic diameter (LV EDD, expressed in mm); left ventricular posterior wall (LVPW, expressed in mm), left atrium diameter (LA, expressed in mm); trans-mitral valve blood flow sampling of early (E) and late (A) peak velocity with E/A ratio, and epicardial fat thickness (EFT). EFT was identified at the interface of external myocardium wall and visceral pericardium was expressed in mm. Patients were examined in the left lateral supine decubitus by an experienced cardiologist. The evaluation was carried out by the same expert cardiologist, to minimize interobserver variability.

### 2.7. Statistical Analysis

The statistical analysis was performed using the software Statistica, version 14 StatSoft Inc. and MedCalc^®^ Statistical Software version 20.111 (MedCalc Software Ltd., Ostend, Belgium; https://www.medcalc.org; accessed on 10 May 2022).

Descriptive statistics (n, mean, SD) was calculated for continuous variables. Frequencies and percentages were presented for categorical variables. Distribution of continuous variables was tested with the Shapiro–Wilk test.

Comparison between groups was made with Student’s *t*-test.

Correlations were corrected for sex, age and BMI and considered statistically significant when *p* < 0.05. The differences between metabolic syndrome and GHD frequencies were evaluated by Chi-squared test. Multiple regression analysis was performed to evaluate independent predictors of LVMi among age, sex, BMI, systolic and diastolic BP, UFDI and AUC GHRH + Arg (ng/mL/hour).

## 3. Results

We enrolled 353 (282 women and 71 men) patients with overweight or obesity. The mean BMI was 38.71 ± 7.88 kg/m^2^ and the mean age was 46.2 ± 12.8 years. General characteristics and anthropometric parameters of our cohort are shown in Appendix A, and metabolic parameters are reported in Appendix A. Echocardiographic parameters are listed in Appendix A.

157 patients (116 females and 41 males)-44.5% of the total—were hypertensive and were effectively treated with 1 or at most 2 antihypertensive drugs. Among these hypertensive patients, 71 were GHD (49.3%) and 86 were non-GHD (41.1%).


GH Secretion


144 patients were diagnosed with GHD, among whom 101 were women. The results of the GH secretion tests are reported in Table 1. In the GHD group, the area under the curve for the GHRH + Arginine test did not statistically differ among men and women, but the GH peak was significantly higher in females than in males (3.44 ± 1.87 vs. 2.65 ± 1.51 ng/mL *p* = 0.016).


Metabolic parameters and body composition


When comparing patients with GHD and patients with normal GH secretion, considered as a control group, differences emerged between the two populations in terms of BMI, metabolic parameters and body composition. The results are shown in Table 2 and Table 3.

The prevalence of metabolic syndrome in male patients diagnosed with GHD was 81%, much higher than in females, where it was 58% (*p* < 0.01).

Metabolic syndrome was more prevalent in the GHD group (Table 3). In particular, linear correlations between GH secretory capacity and metabolic parameters, after adjustment for sex, age and BMI, showed that the AUC of the dynamic test with GHRH + arginine correlated positively with fasting blood glucose, triglycerides, insulin and systolic and diastolic blood pressure values, and negatively with HDL cholesterol values (see Table 4 and Table 5).


Echocardiographic parameters


Interestingly, many echocardiographic parameters were significantly different in GHD and non-GHD patients, notably EFT (*p* < 0.001), left ventricular mass index (LVMi) (*p* < 0.001), E/A ratio (*p* < 0.001) and ejection fraction (*p* < 0.001) (see Table 6).

Dividing our GHD population by gender, EF was higher in female patients compared to men (65.14 ± 4.01 vs. 61.51 ± 7.65%, *p* = 0.002), while E/A did not statistically differ among sexes.

Epicardial fat was significantly higher in males than females (9.76 ± 1.339 mm vs. 8.048 ± 0.717, *p* < 0.001).

The linear regression method also showed that GH secretion had an inverse correlation with left ventricular mass index (r = −0.29, *p* < 0.001) and EFT (r = −0.39, *p* < 0.001) and a direct correlation with E/A ratio (r = 0.26, *p* < 0.001), respectively (see Figure 1, Figure 2 and Figure 3).

A multiple linear regression analysis was performed to assess independent predictors of LVMi and showed that older age and male gender positively correlated with higher LVMi, whilst GH secretory capacity (AUC GHRH + Arginine (ng/mL/hour)) appeared protective for LV hypertrophy (see Table 7).

## 4. Discussion

In recent decades, the cardiovascular risk in patients with hypopituitarism and the role of GH in defining this risk have been studied [12,13,14,15]. GHD may be associated with important cardiovascular risk factors such as altered lipid and glucose levels, high blood pressure, increased inflammatory status and altered body composition, leading to a profile of increased cardiovascular risk [16]. Impairment of the GH/IGF-1 axis appears to be associated with the risk of developing sarcopenic obesity and ectopic fat deposition in the liver [11]. Furthermore, sarcopenic obesity has been shown to be associated with MS and low-grade inflammation in adult subjects [17,18]. The cardiovascular risk is closely related to MS and its defining parameters, which are also common in GHD, namely, increased waist circumference—an expression of visceral obesity—altered glucose levels, hypertension and dyslipidaemia. All these alterations are also often found in patients with obesity.

Patients with panhypopituitarism without GH replacement therapy showed a higher prevalence of MS [19,20,21,22]. One of the strongest predictors of MS in panhypopituitarism is obesity, and its increased prevalence may be associated with the metabolic alteration caused by GH deficiency [23]. The prevalence of MS in patients with GHD is still being debated. In our study, GHD patients showed a significantly higher prevalence of MS (55%), compared to the control group (45%, *p* < 0.0001). These findings appear to be in line with those of other studies [24] and with epidemiological studies [25,26], analyzing the prevalence of GHD and MS in the general population. Some studies did not observe significant differences in MS prevalence between GHD and controls, but nevertheless, some key parameters of MS, especially waist circumference and dyslipidaemia, were significantly higher in patients with GHD [27,28,29,30], although a recent study [31] showed no significant alteration of metabolic risk factors in GHD patients not treated with replacement therapy. On the contrary, we observed higher fasting glucose, higher blood pressure values, higher levels of triglycerides and lower HDL cholesterol and higher BMI in patients with GHD, confirming the higher percentage of MS among GHD patients.

Regarding body composition, we observed that the percentage of lean mass was similar in the GHD group and the control group, but the ratio of ASMM/weight was worse in the GHD group, typifying a condition of relative sarcopenia. The distribution pattern of fat mass was quite different, as the GHD group had a higher visceral fat mass than the controls, confirmed by WC and WHR and by DXA parameters of TF and UFDI.

The ectopic fat deposition was also evidenced by echocardiography as follows: we found a higher EFT in patients with GHD, similarly to other authors [32]. The EF is a metabolically active organ that generates a variety of bioactive molecules with inflammatory properties that could significantly affect cardiac function [33,34]. There is evidence that ultrasound-estimated EFT could be applied as an easy and reliable imaging indicator of ectopic fat and cardiovascular risk.

Furthermore, Ferrante et al. [35] showed a reduction in EFT after 6 months of GH replacement therapy, and the reduction was maintained trend at 12 months. Changes in EFT did not correlate with changes in BMI, WC or metabolic parameters and there was no correlation between baseline EFT and its reduction. In addition, a global improvement in diastolic function was reported, probably due to the reduction in EFT. Considering the EF is a pro-inflammatory tissue, its reduction is a significant achievement in lowering patients’ cardiometabolic risk.

Similarly to other studies, we found significantly higher levels of C-reactive protein in GHD patients, confirming that GHD is related to a pro-inflammatory state, which is a known contributor to cardiovascular damage [14,30,36,37].

However, the effects of GH replacement therapy on metabolic parameters and cardiovascular risk are still being debated. Verhelst et al. [38] found that, in patients with GHD, a replacement treatment did not appear to improve the prevalence of MS. On the contrary, an Italian study reported that GH replacement therapy reduced total and LDL-cholesterol concentrations, but the Framingham cardiovascular risk index increased over time, probably due to the subjects’ increasing age [22]. Patients with MS appear to better respond to GH replacement [39] and other studies have reported changes in body composition, decreased LDL levels and improved BP after GH supplementation [40,41,42,43,44], although not all studies agree on the improvement of all these parameters.

GH plays an important role in maintaining the structure and function of the normal adult heart. The GH/IGF-I axis also interacts with the vascular system and is involved in the regulation of vascular tone [13].

Patients with GHD may present with an altered cardiac mass, but the results are not conclusive. Indeed, adults with childhood-onset GHD have a reduced cardiac mass [45,46], though this is not always observed in adult-onset GHD, which often has a normal cardiac mass [29,45,47,48].

Interestingly, we observed that patients in the GHD group had a significantly higher LVMi.

Our results could be related to the peculiar characteristics of our population, which was suffering from obesity (mean BMI in GHD group 41.7 ± 9.1). To possibly explain the increased myocardial mass, it is important to note that recent studies report increased intra-myocellular fat deposition in obese individuals with blunted GH response [49,50]. However, the data from the majority of studies derive from small populations of normal/slightly overweight patients, or with a childhood-onset GHD. Similarly to our findings, De Gregorio et al. [32] observed that a subgroup of GHD patients had high LVMi and eccentric hypertrophy. This finding is typically observed in patients with obesity, and so we may speculate that ectopic adipose tissue accumulation is the first strong determinant of cardiovascular remodelling in these patients and that GHD may act as an additive factor for increasing cardiometabolic risk by acting on MS elements and exacerbating cardiac alterations. In fact, LVMi has been found to be a particularly important prognostic factor for cardiovascular events [32,51]. Therefore, GHD patients suffering from obesity may represent a population at particularly high risk for cardiovascular events [32].

Diastolic abnormalities may be the first manifestations of heart disease and can be identified with an E/A ratio <1 as type 1 diastolic dysfunction [52].

According to our results, GHD patients had more frequent diastolic dysfunction, consistent with previous studies [48,53]. This diastolic dysfunction, associated with reduced cardiac performance, especially during physical exercise, may be responsible for the reduced exercise capacity [48,54] and reduced quality of life experienced by GHD patients.

The role of GHD in determining diastolic dysfunction has not yet been elucidated, but preclinical studies suggest a potential role of the GH-IGF-1 axis in maintaining myocardial calcium homeostasis, with a direct effect on contractility and a trophic action of the GH on cardiomyocytes [53,54,55,56,57].

On the other hand, the systolic function also appears to be reduced in GHD patients compared to control subjects, but values are still within the normal range, as reported by other studies [43].

GHD and metabolic obesity share harmful actions at the cardiac level. Several studies found that obesity, and in particular visceral obesity, is an independent risk factor for the development of heart failure [58]. GHD represents one of the most common hormone deficiencies observed in patients with chronic heart failure. The mechanisms explaining this phenomenon remain unclear, though chronic liver congestion may reduce IGF-1 levels, thus contributing to the decline of the foremost peripheral GH effects [59]. In our study, independent predictors of greater LVMi, as already reported, were older age, male gender, elevated systolic blood pressure, altered fat distribution and higher BMI, while interestingly, GH secretory capacity appeared protective for LV hypertrophy.

Several studies proved that GH replacement improves these cardiac abnormalities, with a great impact on patients’ quality of life [43,54,60].

A study by Colao et al. showed an improvement in cardiovascular risk parameters, cardiac mass and functional parameters in patients who had received GH replacement [61]. Strikingly, a further impairment of cardiovascular risk parameters, LV performance both at rest and at peak exercise and exercise duration were observed in GHD patients who received complete hormone replacement (where required) except for GH. These findings suggest that 12 months of GH deprivation can exacerbate cardiovascular risk and possibly increase the risk of cardiac accidents [43]. Overall mortality, myocardial infarctions and cerebrovascular events have been observed to be more frequent in patients suffering from hypopituitarism without GH replacement than in the general population [55,62].

A very mild increased risk of developing malignancies during substitutive GH therapy can not be completely excluded, so it is mandatory to propose GH substitutive therapy only to properly diagnosed GHD patients after the collection of a detailed past medical history focused on malignancies, balancing benefits and risks. Furthermore, it is mandatory to properly follow up with the patients and program screening for the most common cancers that patients may develop during treatment.

## 5. Conclusions

In conclusion, GHD can be considered an important cardiometabolic risk factor. All components of the MS, such as fasting glucose, lipid profile, waist circumference and blood pressure, were significantly worse in the GHD patients, confirming that among patients with obesity and overweight, those with GHD have a higher cardiovascular risk than obese and overweight individuals with preserved GH secretory capacity. Alterations in the metabolic phenotype are associated with alterations in cardiac morphology and function.

GHD patients should be screened not only for metabolic impairment but also for cardiac alterations, with the use of echocardiography as a non-invasive and cost-effective tool. GHD alters body composition, leading to increased fat mass and higher visceral adiposity and consequently to a pro-inflammatory status that worsens the prognosis of these patients. Obesity, which is often associated with adult-onset GHD, could influence the cardiac abnormality pattern, configuring a peculiar change in cardiac structure, with increased LVMi instead of the classical reduced cardiac mass. Replacement therapy should be encouraged to avoid worsening metabolic status, cardiac performance and remodelling, with the ultimate goal of reducing cardiovascular adverse events. Targeted clinical trials in patients with obesity and concomitant GHD are needed.

## Figures and Tables

**Figure 1 cells-11-02420-f001:**
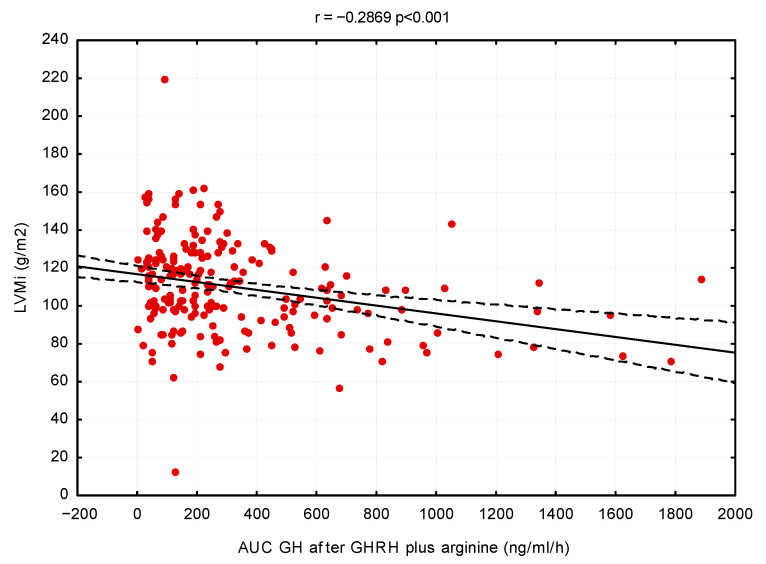
Linear regression between LVMi and GH secretory capacity (AUC. ng/mL/hour)–adjusted for age, sex and BMI.

**Figure 2 cells-11-02420-f002:**
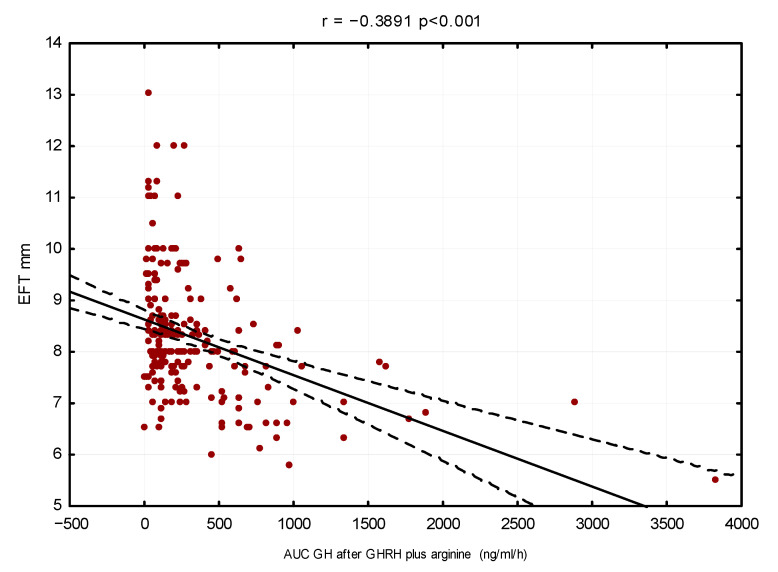
Linear regression between EFT (mm) and GH secretory capacity (AUC. ng/mL/hour)–adjusted for age, sex and BMI.

**Figure 3 cells-11-02420-f003:**
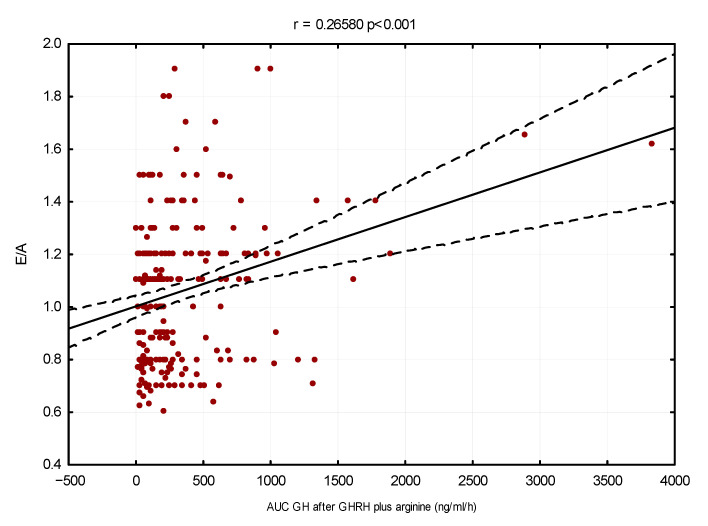
Linear regression between E/A and GH secretory capacity (AUC. ng/mL/hour)–adjusted for age, sex and BMI.

**Table 1 cells-11-02420-t001:** GH secretion parameters in the study population (282 women–71 men).

N (Female/Male)	144 (101/43)	209 (181/28)	
	GHD	Non GHD	*p* Value
GH peak (ng/mL)	3.21 ± 1.80	16.38 ± 11.88	<0.001
AUC GHRH + Arg (ng/mL/hour)	113.72 ± 66.78	510.98 ± 370.09	<0.001

AUC = area under the curve; GHRH + Arg = stimulation test with GH-releasing hormone and arginine; GHD = growth hormone deficiency.

**Table 2 cells-11-02420-t002:** Comparison of general characteristics, anthropometrics and body composition parameters between GHD patients and the control group (patients with normal GH secretion).

	GHD	Non GHD	*p* Value
Age (years)	47.18 ± 11.51	45.58 ± 13.56	NS
BMI (kg/m^2^)	41.74 ± 9.10	36.60 ± 6.12	<0.001
Waist circumference (cm)	127.25 ± 17.97	114.21 ± 14.89	<0.001
Hip circumference (cm)	126.43 ± 15.38	119.78 ± 13.42	<0.001
WHR	0.99 ± 0.10	0.95 ± 0.09	<0.005
FM %	40.25 ± 7.33	40.67 ± 6.32	NS
LM%	59.75 ± 7.33	59,33 ± 632	NS
TF (kg)	20.48 ± 6.13	18.48 ± 5.97	<0.005
TF %	39.24 ± 7.65	38.71 ± 6.61	NS
UFDI	1.99 ± 0.60	1.80 ± 0.54	<0.005
ASMM/weight	0.24 ± 0.04	0.25 ± 0.03	<0.01

BMI = body mass index; WHR = waist-to-hip ratio; FM = fat mass; LM = lean mass; TF = trunk fat; UFDI = upper body fat deposition index; ASSM = appendicular skeletal muscle mass.

**Table 3 cells-11-02420-t003:** Comparison of metabolic parameters between the GHD population and the control group.

	GHD	Non GHD	*p* Value
MS	69.44%	44.23%	<0.0001
Glycemia (mg/dL)	101.44 ± 17.66	95.00 ± 18.57	<0.001
Insulinemia (UI/L)	26.04 ± 20.50	15.83 ± 13.56	<0.001
Glycated haemoglobin %	6.00 ± 0.82	5.48 ± 0.61	<0.001
HOMA-IR	6.68 ± 5.47	3.93 ± 3.85	<0.001
Triglycerides (mg/dL)	151.56 ± 88.48	128.60 ± 117.37	<0.05
HDL (mg/dL)	46.60 ± 12.18	50.83 ± 13.03	<0.005
LDL (mg/dL)	119.90 ± 38.55	118.47 ± 30.10	NS
Total Cholesterol (mg/dL)	195.91 ± 40.97	194.68 ± 34.18	NS
CRP (mg/L)	0.74 ± 0.58	0.60 ± 0.56	<0.05
Systolic BP (mmHg)	132.25 ± 16.56	128.23 ± 18.15	<0.05
Diastolic BP (mmHg)	83.04 ± 11.46	80.55 ± 11.84	<0.05

MS = metabolic syndrome; HOMA-IR = homeostatic model assessment of insulin resistance; HDL = high density lipoproteins; LDL = low density lipoproteins; CRP = C-reactive protein; BP = blood pressure; GHD = growth hormone deficiency.

**Table 4 cells-11-02420-t004:** Frequencies chart of metabolic syndrome and GHD in the population studied. Chi-squared test, *p* < 0.0001.

	Metabolic Syndrome		
GHD	0	1	
0	117 55.5% RT 73.6% CT 33.1% GT	94 44.5% RT 48.5% CT 26.6% GT	211 (59.8%)
1	42 29.6% RT 26.4% CT 11.9% GT	100 70.4% RT 51.5% CT 28.3% GT	142 (40.2%)
	159 (45.0%)	194 (55.0%)	353

**Table 5 cells-11-02420-t005:** Linear correlations between the GH secretory capacity and metabolic parameters (adjusted for sex, age and BMI).

	Correlations between AUC GH after GHRH Plus Arginine (ng/mL/hour) and Metabolic Parameters (Corrected for Sex, Age and BMI)								
	SBP mm Hg	DBP mm Hg	Total Chol mg/dL	LDL Chol mg/dL	HDL Chol mg/dL	TG mg/dL	Glycemia mg/dL	Insulin UI/L	HOMA-IR
AUC GH after GHRH plus arginine (ng/mL/hour)	−0.3371	−0.2831	−0.0359	−0.0612	0.2236	−0.2418	−0.2601	−0.2712	−0.2616
	*p* < 0.001	*p* < 0.001	*p* = 0.563	*p* = 0.325	*p* < 0.001	*p* < 0.001	*p* < 0.001	*p* < 0.001	*p* < 0.001

**Table 6 cells-11-02420-t006:** Comparison of echocardiographic parameters between the GHD population and the control group.

	GHD	Non GHD	*p* Value
EF %	63.91 ± 5.75	65.58 ± 3.60	<0.001
E/A	1.02 ± 0.22	1.10 ± 0.29	<0.001
1 type DD	52%	38%	<0.01
cFS %	38.34 ± 3.85	38.89 ± 3.08	NS
EFT (mm)	8.65 ± 1.27	7.95 ± 1.00	<0.001
LVMI (g/m^2^)	118.32 ± 24.54	105.98 ± 22.52	<0.001
IVS (mm)	11.22 ± 1.54	10.50 ± 1.45	<0.001
LVPW (mm)	10.17 ± 1.04	9.69 ± 1.12	<0.001
LV EDD (mm)	51.45 ± 5.02	48.92 ± 3.64	<0.001
LV ESD (mm)	31.79 ± 4.20	29.70 ± 3.39	<0.001
LA (mm)	40.48 ± 3.32	37.12 ± 4.21	<0.001

EF = ejection fraction; E/A = ratio between E wave and A wave; DD = diastolic dysfunction; cFS = circumferential fractional shortening; EFT = epicardial fat thickness; LVMI = left ventricular mass index; IVS = inter ventricular septal dimension; LVPW = left ventricular posterior wall; LV EDD = left ventricular end diastolic diameter; LV ESD = left ventricular end systolic diameter; LA = left atrium; GHD = growth hormone deficiency.

**Table 7 cells-11-02420-t007:** Multiple linear regression analysis.

Independent Variables	Coefficient	Std. Error	t	*p*	r_partial_	r_semipartial_
(Constant)	52.5253					
AGE (years)	0.7193	0.1227	5.862	<0.0001	0.4061	0.3585
AUC_GH (ng/mL/hour)	−0.01361	0.004778	−2.848	0.0049	−0.2110	0.1742
BMI_(Kg/m^2^)	0.2028	0.2401	0.845	0.3994	0.06391	0.05166
UFDI	−1.1558	2.3892	−0.484	0.6292	−0.03665	0.02959
SBP mmHg	−0.04224	0.1019	−0.415	0.6790	−0.03141	0.02535
SEX (1 = male)	23.7389	4.0405	5.875	<0.0001	0.4069	0.3593

## Data Availability

Data will be made available upon reasonable request to the corresponding author.

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
