# Peer review of "Growth Hormone Secretory Capacity Is Associated with Cardiac Morphology and Function in Overweight and Obese Patients: A Controlled, Cross-Sectional Study"

_cells, 2022, doi:10.3390/cells11152420_

Round 1

Reviewer 1 Report

This is a well-designed study using appropriate clinical and laboratory methods. It proves that obese GH deficient adults are at a greater risk to develop CVD than non GH deficient adults with a similar degree of obesity.

Well written.

Author Response

This is a well-designed study using appropriate clinical and laboratory methods. It proves that obese GH deficient adults are at a greater risk to develop CVD than nonGH deficient adults with a similar degree of obesity.

Well written.

Dear reviewer, thank you very much for your comments, we are glad you appreciated our manuscript.

Reviewer 2 Report

The manuscript by Gangitano et al. sought to investigate the association of GH secretion with metabolic phenotype and the cardiac function and morphology in patients with overweight or obesity. They found that GH secretory capacity was related to metabolic phenotype, body composition, cardiovascular risk factors, and cardiac morphology. The reviewer has the following comments.

1.      Tables 1, 2, and 3 are unnecessary in the main text or could be moved to supplemental data since these are general information about the included patients. The same data was split into GHD and non-GHD and presented again in Tables 5-7.

2.      142 patients were diagnosed with GHD, and the rest of the 353 patients were non-GHD. However, in Table 7, there are 144 patients in GHD and 208 patients in non-GHD. The numbers are not consistent.

3.      The majority of patients with GHD are female (101 vs. 41). Are there any sex differences in GH secretion capacity and its association with cardiac function and morphology?

4.      Line 220-223, “A multiple linear regression analysis…showed that older age and male gender positively correlated with higher LVMi, .... appeared protective for LV hypertrophy (Table 10)”. However, in the last row (SEX) of Table 10, it indicates "female". Which one is correct?

5.      The Tables should be better formatted and presented. Such as adding borders and lines to separate titles, data, and abbreviations. 

Author Response

The manuscript by Gangitano et al. sought to investigate the association of GH secretion with metabolic phenotype and the cardiac function and morphology in patients with overweight or obesity. They found that GH secretory capacity was related to metabolic phenotype, body composition, cardiovascular risk factors, and cardiac morphology. The reviewer has the following comments.

  1. Tables 1, 2, and 3 are unnecessary in the main text or could be moved to supplemental data since these are general information about the included patients. The same data was split into GHD and non-GHD and presented again in Tables 5-7.

    Dear Reviewer, thanks for your comment. We moved the tables you mentioned as supplementary data.

  2. 142 patients were diagnosed with GHD, and the rest of the 353 patients were non-GHD. However, in Table 7, there are 144 patients in GHD and 208 patients in non-GHD. The numbers are not consistent                  Thanks a lot for noticing it. We revised the data and corrected the oversight accordingly 
  3. The majority of patients with GHD are female (101 vs. 41). Are there any sex differences in GH secretion capacity and its association with cardiac function and morphology? 

    Thanks for the question. We specifically looked for it,and observed that the area under the curve for GHRH+Arginine test did not statistically differ among men and women, but the GH peak was significantly higher in females than males (3.44±1.87 mg/ml vs 2.65±1.51 p=0.016) in the GHD group.

    The EF was higher in female patients (65.14±4.01 vs 61.51±7.65% , p=0.002), while E/A did not statistically differ among sex.

    Epicardial fat was significantly higher in males than females (9.76±1.339 mm vs 8.048±0.717, p<0.001).

  4. Line 220-223, “A multiple linear regression analysis…showed that older age and malegender positively correlated with higher LVMi, .... appeared protective for LV hypertrophy (Table 10)”. However, in the last row (SEX) of Table 10, it indicates "female". Which one is correct? 

    Thanks, we revised it and corrected it.

  5. The Tables should be better formatted and presented. Such as adding borders and lines to separate titles, data, and abbreviations.  

    Thanks for your comment. We ameliorated the formatting of the tables to make them more easily readable.

Reviewer 3 Report

This cross-sectional study presents a high clinical relevance. The association between the deficiency in Growth hormone and conditions related to Metabolic syndrome and with cardiac morphology and function are studied in patients. This is a valuable study, but I think that some aspects could be improved.

1 – Data included in Tables 1, 4 and 5, besides global values should present data discriminated by sex. This is relevant considering the high-incidence of GHD in women. In the tables 2,3, 6 and 7 legends, at least the number of women and men in GHD and non-GHD groups should be indicated.

2 – Did the authors find differences between women and men in the prevalence of MS in GHD? It should be mentioned in the text.

3 – It will be, as well, important to indicate if patients receive medication to control hypertension, diabetes, dyslipidemia, etc.

4 – The results should be divided in subsections.

5 – The risk of promoting cancer development in obese patients after GH replacement should be discussed. While mitigating CVD risk, malignancies could arise.

Minor corrections.

6 – Lines 84-85.  It is not necessary to repeat the conditions associated with  GHD

7 – Correction of sentences in lines 270, 291 and 309.

Author Response

This cross-sectional study presents a high clinical relevance. The association between the deficiency in Growth hormone and conditions related to Metabolic syndrome and with cardiac morphology and function are studied in patients. This is a valuable study, but I think that some aspects could be improved.

1 – Data included in Tables 1, 4 and 5, besides global values should present data discriminated by sex. This is relevant considering the high-incidence of GHD in women. In the tables 2,3, 6 and 7 legends, at least the number of women and men in GHD and non-GHD groups should be indicated. 

Thanks for your comment. We moved some tables in the supplementary data as requested by another reviewer, but inserted this gender related information in the text.

2 – Did the authors find differences between women and men in the prevalence of MS in GHD? It should be mentioned in the text. 

Thanks for asking this interesting point, we inserted it as well. 

3 – It will be, as well, important to indicate if patients receive medication to control hypertension, diabetes, dyslipidemia, etc.  

Most patients, being obese, had significant comorbidities and received medications to control hypothyroidism, diabetes mellitus type 2, dyslipidemia , hypertension, gastritis and depression when required. We insert this sentence in the section 2.“Materials and Methods”

4 – The results should be divided in subsections.                                                 Thanks, we did it as suggested

5 – The risk of promoting cancer development in obese patients after GH replacement should be discussed. While mitigating CVD risk, malignancies could arise.  

Thanks for pointing this. We added it in the discussion.

Minor corrections.

6 – Lines 84-85.  It is not necessary to repeat the conditions associated with GHD Thanks, done

7 – Correction of sentences in lines 270, 291 and 309.

Thanks, done